# In Silico Investigation of the Human GTP Cyclohydrolase 1 Enzyme Reveals the Potential of Drug Repurposing Approaches towards the Discovery of Effective BH_4_ Therapeutics

**DOI:** 10.3390/ijms24021210

**Published:** 2023-01-07

**Authors:** Dania Hussein

**Affiliations:** Department of Pharmacology and Toxicology, College of Clinical Pharmacy, Imam Abdulrahman bin Faisal University, Khobar 31441, Saudi Arabia; dahussein@iau.edu.sa; Tel.: +96-6506-271-501

**Keywords:** molecular dynamics, drug repurposing, GTPCH1, BH_4_, inflammation, pain

## Abstract

The GTP cyclohydrolase 1 enzyme (GTPCH1) is the rate-limiting enzyme of the tetrahydrobiopterin (BH_4_) biosynthetic pathway. Physiologically, BH_4_ plays a crucial role as an essential cofactor for the production of catecholamine neurotransmitters, including epinephrine, norepinephrine and dopamine, as well as the gaseous signaling molecule, nitric oxide. Pathological levels of the cofactor have been reported in a number of disease states, such as inflammatory conditions, neuropathic pain and cancer. Targeting the GTPCH1 enzyme has great potential in the management of a number of disease pathologies associated with dysregulated BH_4_ physiology. This study is an in silico investigation of the human GTPCH1 enzyme using virtual screening and molecular dynamic simulation to identify molecules that can be repurposed to therapeutically target the enzyme. A three-tier molecular docking protocol was employed in the virtual screening of a comprehensive library of over 7000 approved medications and nutraceuticals in order to identify hit compounds capable of binding to the GTPCH1 binding pocket with the highest affinity. Hit compounds were further verified by molecular dynamic simulation studies to provide a detailed insight regarding the stability and nature of the binding interaction. In this study, we identify a number of drugs and natural compounds with recognized anti-inflammatory, analgesic and cytotoxic effects, including the aminosalicylate olsalazine, the antiepileptic phenytoin catechol, and the phlorotannins phlorofucofuroeckol and eckol. Our results suggest that the therapeutic and clinical effects of hit compounds may be partially attributed to the inhibition of the GTPCH1 enzyme. Notably, this study offers an understanding of the off-target effects of a number of compounds and advocates the potential role of aminosalicylates in the regulation of BH_4_ production in inflammatory disease states. It highlights an in silico drug repurposing approach to identify a potential means of safely targeting the BH_4_ biosynthetic pathway using established therapeutic agents.

## 1. Introduction

GTP cyclohydrolase 1 (GTPCH1) catalyzes the rate-limiting step in the biosynthesis of the pteridin (6R)-L-erythro-5,6,7,8-tetrahydrobiopterin (BH_4_), an essential cofactor for a number of key enzymes, including the three isoforms of nitric oxide synthase, aromatic amino acid hydroxylases and glyceraldehyde monooxygenase [1,2]. BH_4_ is essential for the synthesis of serotonin, epinephrine, norepinephrine, dopamine and nitric oxide, as well as the metabolism of glycerol-ethers, and as such plays an essential role in the physiological regulation of hemodynamics and neurotransmission [3].

The crystal structure of the human GTPCH1 enzyme has been solved (see Appendix A for a summary of the solved GTPCH1 crystal structures and co-crystalized ligands). The GTPCH1 enzyme is a homodecamer widely expressed within human tissues [4], and enzyme activity is regulated by allosteric means via association with the regulatory binding protein GTP cyclohydrolase 1 feedback regulatory protein (GFRP) [5]. The regulation of enzyme activity and expression is essential to ensure the strict control of BH_4_ levels as the cofactor plays a vital role in a number of physiological processes [6]. GTPCH1 is the first and the rate-limiting enzyme in the three step pathway towards the production of BH_4_. It converts the substrate guanosine triphosphate (GTP) to its intermediate 7,8-dihydroneopterin triphosphate, which is subsequently converted to BH_4_ by 6-pyruvoyl-tetrahydrobipterin synthase (6-PTS) and sepiapterin reductase (SR) [7]. Figure 1 depicts the biosynthetic pathway.

While GTPCH1 plays a well-documented role within the setting of both cardiovascular and neurological diseases [3,8,9], it has also garnered significant interest regarding the role that it may have in pain pathways [10]. In a hallmark study, the enzyme was identified as a key modulator in peripheral neuropathic and inflammatory pain settings [11]. Tregeder et al. first observed that enzyme expression and consequent levels of BH_4_ were increased in sensory nerves and ganglionic tissue following axonal injury and peripheral inflammation. Furthermore, in animal models, inhibiting GTPCH1 activity were shown to reduce neuropathic and inflammatory pain responses, whereas this was increased upon the direct administration of BH_4_ [11,12,13]. Finally, it has been reported that a GTPCH1 genetic haplotype in humans was associated with reduced pain sensitivity [11]. Consequently, further studies have identified a number of single nucleotide polymorphisms within the GTPCH1 gene that are found to be associated with a pain-resistant phenotypes [14,15].

A number of recent studies have implicated the GTPCH1/BH_4_ axis in the pathogenesis of cancer. The role of the GTPCH1 enzyme in the context of cancer and tumor growth is complex, yet in multiple studies, GTPCH1 expression appears to be upregulated in tumors and various cancer cell lines [16]. A number of in vivo and in vitro experiments have shown that GCTPH1 inhibition attenuated tumor growth in triple-negative breast cancer [17], glioblastoma [18] and esophageal squamous cell carcinoma [19]. In colorectal cancer, the inhibition of the GTPCH1/BH_4_ axis appears to be a promising pathway in synergistically assisting non-apoptotic cell death when used in combined treatment with erastin, a novel ferroptosis inhibitor [20,21].

Inhibiting the enzymes involved in BH_4_ biosynthesis represents a novel means of therapeutically targeting inflammatory and neuropathic pain pathways and in suppressing cancer cell growth in susceptible cell lines. Identifying agents that are capable of binding and inhibiting GTPCH1, the rate-limiting enzyme of the pathway, is highly promising for the discovery of novel therapies for pain and cancer. Drug repurposing offers great potential in the utilization of approved therapeutics to identify novel targets for the management of different disease conditions. In silico and computational modeling platforms that visualize and identify ligand–target interactions represent a powerful and efficient approach towards successful drug discovery and repurposing [22,23].

This study aims to conduct a detailed in silico investigation into the human GTPCH1 enzyme using virtual screening and molecular dynamic simulation studies to identify compounds that can be repurposed in order to target the enzyme therapeutically. A comprehensive virtual screening approach is used to identify approved drugs and natural compounds that may be potentially repurposed to inhibit the GTPCH1 enzyme. A three-tier molecular docking protocol enabled the identification of hit molecules capable of binding to the GTPCH1 binding pocket with the highest binding capacity. Furthermore, the interaction profile of all hit compounds is verified by molecular dynamic simulation, providing a detailed insight into the stability and nature of the binding interaction.

## 2. Materials and Methods

### 2.1. Materials and Computational Tools

Computational tools: computational simulations were carried out using the Maestro graphical user interface of Schrödinger [24]. For molecular dynamic simulation studies, a desktop workstation with Linux Ubuntu operating system was used and supported with a RTX 5000 graphics card.

### 2.2. Crystal Structures and Protein Preparation

The human GTPCH1 crystal structure was retrieved from protein databank PDB: 6Z86. Selected structures were solved by X-ray crystallography with a resolution of 2.21 Å. All crystal structures were prepared using the Protein Preparation Wizard tool. Structure preparation and minimization was carried out at a pH 7.4 with corrected ionization states. Polar hydrogens were added and non-essential water molecules were removed. The entire structure was minimized and optimized with the OPLS3 force field, and RMSD default values of 0.30 Å for non-hydrogen atoms were used.

### 2.3. Virtual Databases and Ligand Library Preparation

A comprehensive library of approved drugs and natural compounds with verified physiological activity was developed. A library of FDA-approved medications was retrieved from the DrugBank database [25]; a total of 1100 drugs were downloaded in SDF format, along with a library of approved nutraceuticals comprising 72 compounds. Furthermore, a library of drugs approved worldwide (non-FDA) was also retrieved from the DrugBank database, containing a total of 3400 downloaded SDF structures for medications. Additionally, a total of 5500 FDA and worldwide approved medications, along with 1537 natural compounds with verified in vivo activity, was retrieved from the Zinc database [26]. The final library of therapeutic drugs and nutraceuticals was comprised of a total of 7074 compounds (excluding redundant structures). For ligand preparation, Maestro’s Ligprep was used. The downloaded SDF structures were converted to a 3D Maestro format. Optimal chirality and ionization states were determined at a physiological pH of 7.4 and the geometries were optimized using an OPLS3 force field. The final 3D conformations were used as the initial input structures for docking.

### 2.4. Binding Pocket Determination and Docking Studies

For docking studies, a single identified binding pocket located between the interface of three associated monomers of the homodecameric protein was used in a high throughput screen of approved drugs and nutraceuticals. The Sitemap tool was used to analyze the experimentally verified binding pocket. Key binding resides of the pocket were identified by analysis of the ligand binding profile in solved crystal structures. A docking grid was generated using glide software and identified by selecting the ligand-binding pocket of the crystal structures co-crystallized with the bound natural ligands and inhibitor. As a control, the identified co-crystallized ligands were docked using the same protocol in order to verify the accuracy of the docking poses and interactions determined using root mean square deviation calculations (RMSD). The receptor grids were generated using a 1.00 van der Waals (vdw) radius scaling factor and a 0.25 partial charge cutoff, with the receptor grids centered on the bound ligand. Binding sites were enclosed within the grid box using default parameters and without constraints. Docking was repeated and verified using three screening settings. All compounds were screened using a high throughput docking setting and the top 200 compounds with the highest binding scores were then selected for standard precision docking; of these verified hits, the top 80 compounds were further verified using extra precision docking settings. The ligands were docked using the extra precision mode (XP) without any constraints, a 0.80 van der Waals (vdw) radius scaling factor and a 0.15 partial charge cutoff. The docking protocol employed allowed for the flexibility of the residues surrounding the ligand-binding pocket. GlideScore was implemented in Glide and was used to estimate the binding affinity and rank the ligands. The XP Pose Rank was used to select the best docked pose for each ligand. The final list of thrice verified compounds was then analyzed in detail based on binding scores and a detailed study of all binding interactions.

### 2.5. Molecular Dynamics Simulation Studies

Molecular dynamic (MD) simulation was carried out using the Desmond Module on Schrödinger’s Maestro platform. The docked complex was first minimized using the Protein Preparation Wizard and the minimized complex was prepared for MD simulation using the system builder application of Desmond. The MD simulation environment generated contained a water-based solvent system: the TIP3P water model. An orthorhombic simulation box with a 10 Å buffer parameter from the protein surface was generated, the entire system was neutralized by calculating and adding the required number of counter ions and 0.15 M NaCl in order to attain isosmotic conditions. MD simulation was carried out at a temperature and atmospheric pressure of 300 K and 1.013 bar, respectively. The simulation was run for a total of 100 nanoseconds (a total of 1000 frames were saved in order to compile the trajectory). Analysis was run and results presented using the simulation interaction diagram tool of Desmond.

## 3. Results

### 3.1. Docking Studies

Docking studies to identify ligands that bind to the GTPCH1-binding pocket involved screening a total of over 7000 approved medications and natural compounds. The root mean square deviation (RMSD) was determined for the docked ligand, the inhibitory substrate 7-deaza-GTP, which was compared to its co-crystallized conformation. RMSD calculation revealed an excellent level of equivalence; RMSD = 1.06 Å and a nearly identical binding profile and pose were noted for the docked ligand compared to the co-crystalized conformation. Interactions with the key residues of the binding pocket were studied in detail. Figure 2 depicts the docked inhibitor and highlights critical interactions with key residues of the binding pocket. Sitemap analysis of the binding pocket revealed a SiteScore of 1.02, a hydrophilic score of 1.08, a hydrophobic score of 0.25 and a Dscore of 1.05. With the docking protocol thus verified, a virtual screen of the comprehensive library was conducted in triplicate as described in detail in the methodology. The three-tier docking protocol provided information on the binding affinity and relative strength and orientation of the ligand-binding interaction. Furthermore, the ligand-binding profiles were compared with those of the inhibitory control compound. The compounds that showed the most favorable binding profile and the best binding scores are shortlisted in Table 1 (2D orientation and binding details for all 15 compounds can be found in the Appendix A).

### 3.2. MD Simulation Results

The top hit molecules were selected based on a detailed visual analysis of the ligand–protein interaction profile. A hit was identified as any compound that had a favorable binding score and binding profile with key residues in the binding pocket. A total of 15 ligands were selected for molecular dynamic simulation studies. Those selected were shown, upon the investigation of the optimal 2D and 3D docking positions, to interact with key residues within the binding pocket. MD simulation studies were essential to validate the stability and strength of the binding interaction over a period of 100 nano seconds. Of the 15 ligands used in MD simulation studies, 6 exhibited a robust and stable binding pattern within the binding pocket of the GTPCH1 enzyme. RMSD plot analysis was used to measure the average displacement of atoms for a particular frame with respect to a reference frame. The analysis revealed a stable binding profile, as indicated by an RMSD fluctuation range within 2 Å of the protein backbone and the ligand throughout the simulation period. Figure 3 shows the RMSD plots for the top six ligands expressing the most favorable binding profiles with key residues in the binding pocket (RMSD fluctuation range 2–4 Å). Figure 4 shows the six hit ligands docked within the binding pocket. The detailed interaction profile and 3D binding frames for individual ligands olsalazine, phenytoin catechol and phlorofucofuroeckol are depicted in Figure 5, Figure 6, Figure 7 and Figure 8.

Olsalazine exhibits one of the most favorable binding profiles, interacting with a number of key residues of the GTPCH1-binding pocket, as shown in Figure 5. Notably, the key binding residues Arg 97 and His 144 formed a stable hydrogen and ionic bond, respectively, with the ligand for 80–100% of the simulation period. Other significant interactions were noted with Lys 93 and Leu 94, as well as Cys 141 and Cys 212, where stable ionic and hydrogen bonds were observed for the duration of the simulation period. All significant interactions are represented in Figure 5A,B, which highlights the residues with the strongest stable ligand interactions throughout the entire simulation period. Figure 5C depicts all significant interactions displayed by the ligand and interacting residues occurring for over 30% of the simulation period.

The MD simulation results for phlorofucofuroeckol in Figure 6 show strong interactions with the residues of the binding pocket. The key binding residues Gln 182 and Glu 183 formed hydrogen bonds with phlorofucofuroeckol, which remained stable throughout the majority of the simulation period. Hydrogen bonds at Gln 89, Asp 119 and Ser 166 also account for the stability of the interaction within the binding pocket. Significant water–bridge interactions were also observed with residues Lys 93, Asp 119 and Ser 166.

Figure 7 shows the binding profile of phenytoin catechol with residues of the GTPCH1-binding pocket. The key binding residues Arg 97, His 144 and Lys 167 formed hydrogen bond interactions with phenytoin catechol. Water–bridge interactions were observed with residues Arg 170 and Arg 97. Significant hydrogen bonds were also observed with other binding residues, namely His 143 and Ser 166. The simulated binding profile suggests strong and stable interactions within the GTPCH1-binding pocket. The final simulation binding frames of the top three hit compounds within the binding pocket are shown in Figure 8, which highlights the most important interactions between key residues in the binding pocket.

## 4. Discussion

The GTP cyclohydrolase 1 enzyme is the rate-limiting enzyme of the tetrahydrobiopterin synthetic pathway [7]. BH_4_ is an essential cofactor for a number of enzymes, including nitric oxide synthases, aromatic amino acid hydroxylases and alkylglycerol monooxygenase [3]. BH_4_-dependent enzymes are crucial for the biosynthesis of critical bioactive molecules and neurotransmitters, such as nitric oxide, serotonin and catecholamines [1,2]. Regulating pathophysiological levels of BH_4_ represents an attractive target for the treatment of disease. Studies have implicated lower levels of BH_4_ in endothelial dysfunction, which is characteristic of cardiovascular disease pathologies. While higher levels of BH_4_ may play a critical role in the pathophysiology of pain and inflammatory pathways [3,10]. Recent studies have highlighted the role of the BH_4_ biosynthetic pathway in regulating cancer cell growth. The findings of these studies are often conflicting; one such study cites the benefit of increasing levels of BH_4_ in order to promote antitumor effects via T cell activation [27]. Yet, other reports have shown elevated BH_4_ levels and GTPCH1 expression to be associated with enhanced gastric cancer cell proliferation [28], increased proliferation of acute lymphoblastic leukemias and lymphomas [29] and the promotion of tumor angiogenesis [30,31]. The complexity of the BH_4_ pathway in cancer cell growth remains to be fully elucidated. Nevertheless, it is evident from a number of preclinical and preliminary clinical studies that the BH_4_ pathway may hold great therapeutic potential [32,33,34]. To date, however, while there appears to be great success in targeting the BH_4_ pathway experimentally, translating such effects to a clinical setting and the development of novel therapeutics, has had only limited success.

Progress in the development of effective BH_4_ therapeutics is limited for a number of reasons. BH_4_, while physiologically potent, is susceptible to environmental oxidation and gastrointestinal degradation, thus limiting its viability as an orally active medication for the management of chronic cardiovascular conditions [3]. Targeting the biosynthetic pathway may present an attractive option, particularly in disease states associated with higher BH_4_ levels [6]. Targeting the rate-limiting enzyme GTPCH1 has had limited success. The challenge here arises from the apparent inaccessibility of the active site within the barrel-like structure of the homodecamer in which it is embedded [35,36]. Despite this, active site inhibitors have been reported in the literature [36,37,38]. The GTPCH1 enzyme structure has been successfully solved using the inactive substrate and active site inhibitor, 7-deazaGTP [38,39,40]. Additionally, 2,4-diamino-6-hydroxypyrimidine (DAHP), which acts as a competitive active site inhibitor (IC50: 300 μM) [36,41], has been shown to produce analgesia in rats [42]. These studies have helped to describe a molecular inhibitory binding pattern and have demonstrated the success of targeting the enzyme within an in vivo setting. While targeting GTPCH1 active sites embedded within the barrel-like structure of the enzyme is challenging, GTP remains accessible at picomolar concentrations [37]. Studies have suggested that the oligomeric structure of the enzyme may be subject to a substrate/zinc-dependent dissociation–reassociation pattern [43,44]. The dissociation and reassociation of oligomeric structures has been shown in a number of physiologically active enzymes to be essential for the allosteric regulation of activity [45]. The dynamic properties of such oligomers may allow for sufficient structural plasticity to enable active site accessibility [44,45].

Drug repurposing is an approach used to discover novel applications for existing therapeutics, and it is extremely useful for identifying and utilizing the off-target effects of clinically approved medications. A hybrid screening study identified sulfasalazine as an inhibitor of sepiapterin reductase, an enzyme downstream of GTPCH1 within the BH_4_ biosynthetic pathway (Figure 1) [46]. An excess of BH_4_ is implicated in inflammatory conditions and pain pathways [11]. The upregulation of GTPCH1 enzyme expression and activity is observed in both inflammatory tissue and macrophages [47]. Sulfasalazine is a drug used in the treatment of irritable bowel syndrome and rheumatoid arthritis, yet its mechanism of action remains incompletely understood [48]. A number of in vitro, cell-based and animal studies have determined that the therapeutic effects of sulfasalazine may in fact be partially attributed to the inhibition of BH_4_ biosynthesis [49]. Enzyme assays have shown that sulfasalazine and its active metabolite sulfapyridine are potent inhibitors of SPR [46,49]. Furthermore, cell-based studies have shown that the consequence of SPR inhibition by sulfasalazine is a reduction in BH_4_ levels [46]. Finally, in vivo studies have shown that a reduction of BH_4_ levels is consistent with a higher tolerance to painful stimuli in the murine inflammatory models [42,50]. There is evidence in the literature that supports the precedent that targeting the BH_4_ biosynthetic pathway using repurposed drugs may be a viable approach for the management of chronic inflammatory disease conditions.

In our study, we identified a number of drugs that bind to the GTPCH1 active site using molecular modeling platforms. In silico approaches that visualize and identify ligand target interactions are a powerful and efficient means towards successful drug development. In this study, hit molecules identified from virtual screening were selected based on a detailed analysis of their interaction profiles. A comprehensive screening library of over 7000 compounds was compiled, comprising both FDA- and worldwide-approved drugs and nutraceuticals, in addition to those natural products with established in vivo activity (Zinc) [51] (Drugbank) [25]. Drugs and compounds found to show a favorable binding profile were shortlisted for further study. Interestingly, we identified a number of drugs and natural compounds that expressed anti-inflammatory, analgesic and cytotoxic effects. The top 15 hit compounds with the highest affinity scores and with favorable interaction profiles in the binding pocket were selected for further molecular dynamic simulation studies.

A 100 nanosecond MD simulation was run for each of the 15 hit ligands, and 6 of these ligands, namely olsalazine, phenytoin catechol, phlorofucofuroeckol, inosine, eckol and valganciclovir, expressed a stable and robust binding profile (the top 3 with a RMSD < 2 Å). Notably, all six ligands were able to bind with key residues within the binding pocket with high affinity (−7 to −10 kcal/mol). The detailed binding data for olsalazine, phenytoin catechol and phlorofucofuroeckol are depicted in Figure 5, Figure 6 and Figure 7, respectively, and reveal interactions with a number of key residues, including Lys 93, Arg 97, His 144 and Ser 166 (the binding profiles of the remaining hit ligands may be found in the Appendix A). Strong zinc-associated ionic interactions account for the stable binding profile of olsalazine within the binding pocket throughout the 100 nanosecond simulation period. Olsalazine, such as sulfasalazine, is an aminosalicylate anti-inflammatory agent used in the treatment of inflammatory bowel disease and ulcerative colitis [52], yet its role as an anti-inflammatory agent is not completely understood. Its anti-inflammatory activity is attributed to the formation of the active metabolite 5-aminosalicylate (5-ASA/mesalamine) [52]. Our studies suggest that the anti-inflammatory effects of olsalazine may possibly be a result of its interaction with the BH_4_ biosynthetic pathway. Olsalazine is shown to form strong and stable binding interactions in the GTPCH1-active site, mimicking elements of an interaction pattern observed by the inhibitory substrate. Furthermore, its active metabolite 5-ASA has been shown experimentally to inhibit SPR the subsequent enzyme in the BH_4_ biosynthetic pathways [46]. The consequence of enzyme inhibition is a reduction in overall BH_4_ production and decreased inflammatory swelling and pain hypersensitivity in vivo [11]. Our findings support both experimental and clinical observations in the literature that implicate the novel role that aminosalicylate drugs may play in targeting the BH_4_ biosynthetic pathway in inflammatory diseases, such as IBS and rheumatoid arthritis.

Interestingly, virtual screening also identified a number of other hit drugs with off-target effects that may be explained by their proposed interaction with GTPCH1. Hyperkinesia and locomotor disorders have been reported with the prolonged use of the parent drug phenytoin, and further investigations are warranted to determine whether such effects may be due to the impairment of GTPCH1 activity by its metabolite phenytoin catechol. It is well reported that BH_4_ deficient conditions, such as BH_4_-dependent congenital hyperphenylalaninemia and dopa-responsive dystonia, express phenotypes of variable severity, ranging from poor motor control, hypo/hypertonia to Parkinsonism [9,53].

Recently, studies have investigated the BH_4_ pathway as a potential target for cancer treatment. Virtual screening of approved medications and natural bioactive compounds has shortlisted a number of hits which express experimentally verified cytotoxic activity. The phlorotannins eckol and phlorofucofuroeckol have both been shown to express cytotoxic effects in a number of different cancer cell lines [54,55]. Additionally, the brown algae derivatives, eckol and phlorofucofuroeckol have been shown to exhibit anti-inflammatory activity [56,57]. Further studies are required to determine whether their activity profile may be in part a result of the manipulation of GTPCH1 activity.

## 5. Conclusions

Targeting the BH_4_ biosynthetic pathway holds great therapeutic potential for a number of diseases. GTPCH1, the committing and rate-limiting enzyme for BH_4_ biosynthesis, represents an attractive target for drug development. Yet, there is limited success in the development of effective medications that can safely modulate enzyme activity. In our study, we utilized an in silico drug repurposing approach to understand the off-target effects of approved drugs and nutraceuticals and to identify a potential means of safely targeting the BH_4_ biosynthetic pathway using established therapeutic agents. Our studies support a number of experimental findings that suggest that aminosalicylate drugs, such as olsalazine, may partially attribute their anti-inflammatory effects to inhibition of BH_4_ production. In addition, the unknown cytotoxic activity of naturally derived phlorotannins may be partially attributed to GTPCH1-inhibition. While further studies are recommended to verify such effects experimentally, this is the first study to examine the promising approach of targeting the human GTPCH1 enzyme using established therapeutics in a safe and effective manner.

## Figures and Tables

**Figure 1 ijms-24-01210-f001:**
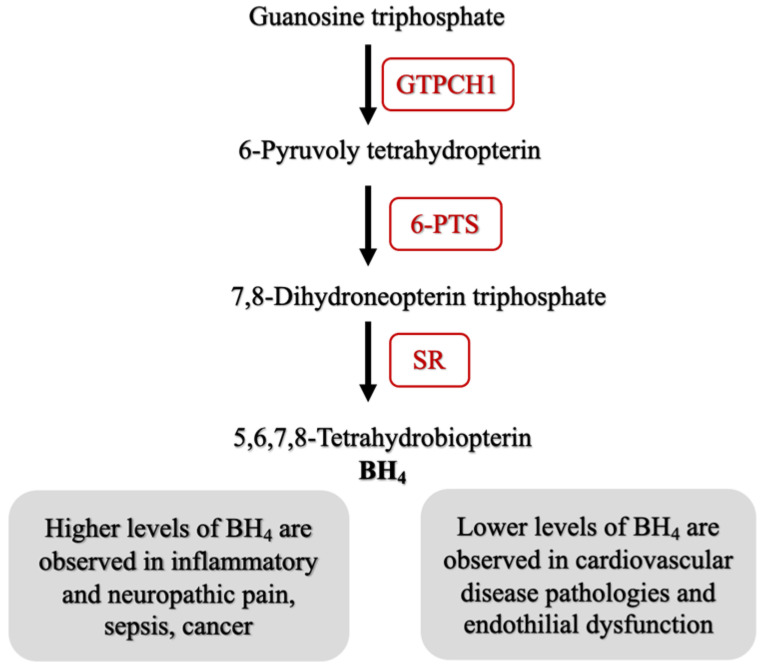
Summary of the BH_4_ biosynthetic pathway and the consequences of the altered physiological levels of the cofactor. GTPCH1: GTP cyclohydrolase 1; 6-PTS: 6-pyruvoyl tetrahydrobiopterin synthase; SR: sepiapterin reductase.

**Figure 2 ijms-24-01210-f002:**
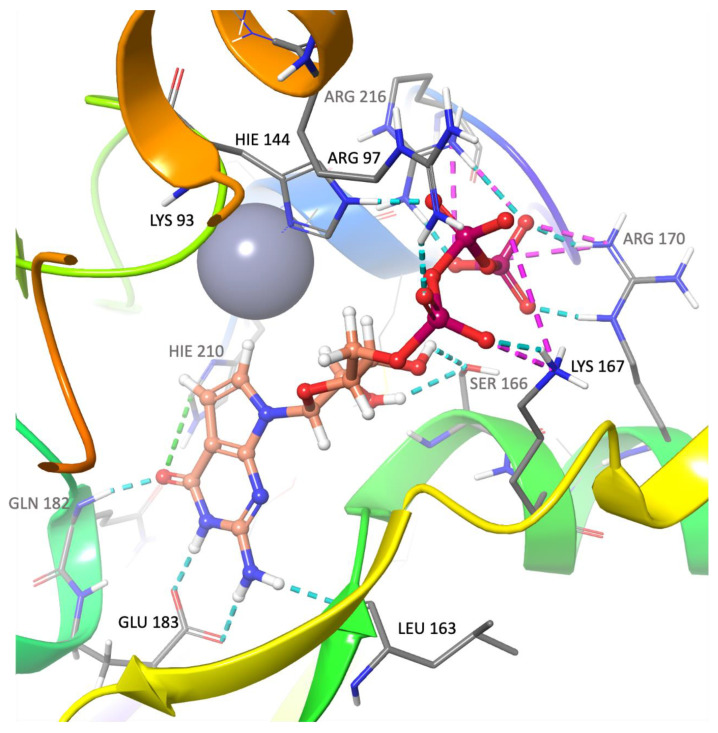
7-Deaza-GTP docked in the GTPCH1-binding pocket (−14.94 kcal/mol). All bonds are depicted as dotted lines: hydrogen bond—blue; salt bridges—pink; aromatic H-bond—green; pi–pi stacking—orange; pi-cation—red. Binding residues are annotated, and the zinc ion is depicted as a grey sphere.

**Figure 3 ijms-24-01210-f003:**
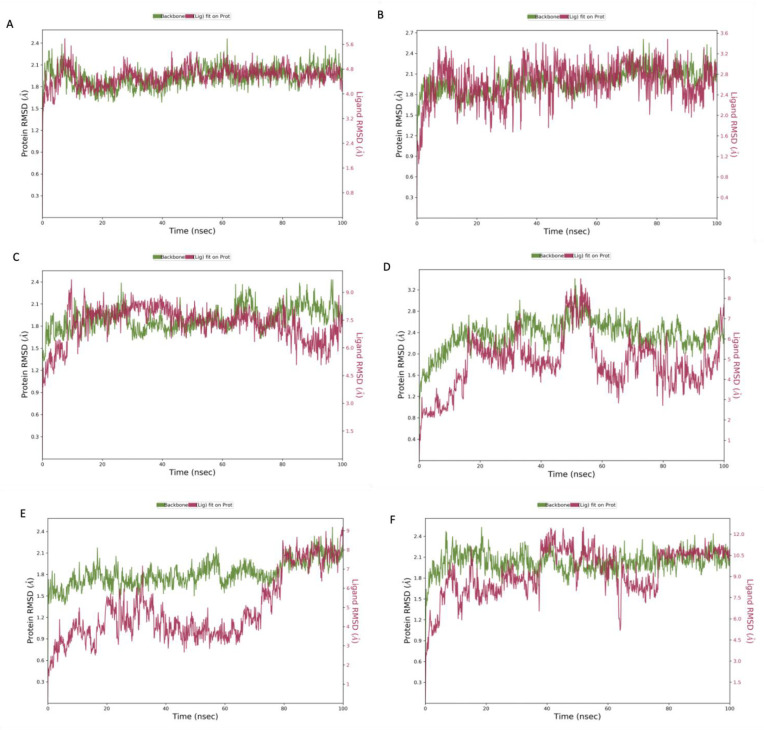
Root mean square deviation (RMSD) graphs for the top hit compounds: (**A**) olsalazine, (**B**) phenytoin catechol, (**C**) phlorofucofuroeckol, (**D**) inosine, (**E**) eckol and (**F**) valganciclovir. The green graph shows the fluctuations in the protein backbone from the initial reference point, while the red shows the ligand fluctuations. The RMSD profile of the ligand is with respect to its initial fit to the protein binding pocket indicates that all ligands did not fluctuate beyond a 2–4 Å range for the majority if the simulation period.

**Figure 4 ijms-24-01210-f004:**
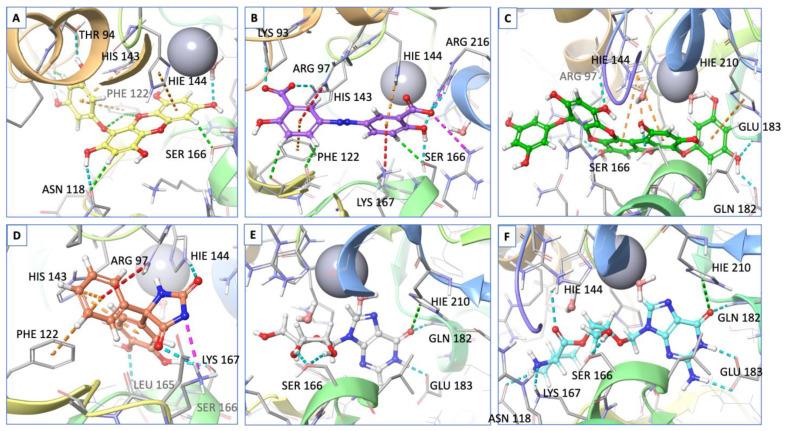
All six hit compounds docked in the GTPCH1-binding pocket: (**A**) Eckol, (**B**) olsalazine, (**C**) phlorofucofuroeckol, (**D**) phenytoin catechol, (**E**) inosine, (**F**) valganciclovir. All bonds are depicted as dotted lines: hydrogen bond—blue; salt bridges—pink; aromatic H-bond—green; pi–pi stacking—orange; pi-cation—red. Binding residues are annotated, and the zinc ion is depicted as a grey sphere.

**Figure 5 ijms-24-01210-f005:**
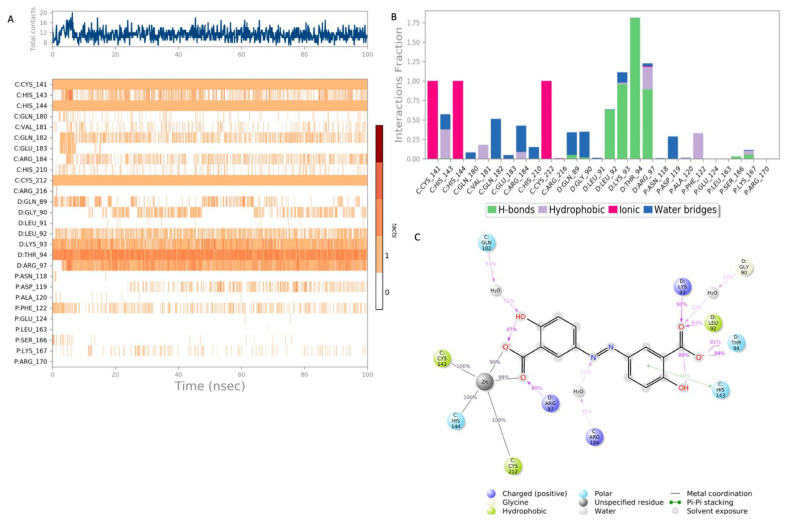
Interaction diagram of olsalazine with the GTPCH1 binding pocket. (**A**) Interaction of olsalazine with residues in each trajectory frame. The depth of color indicating the higher the interaction with contact residues; (**B**) the protein–ligand contacts showing the bonding interaction fractions and the nature of the interactions; (**C**) graphical 2D illustration of olsalazine interacting with the protein residues during MD simulation. Interactions shown occurred over 30% of the simulation time. C—chain C of the GTPCH1 binding pocket; D—chain D of the binding pocket.

**Figure 6 ijms-24-01210-f006:**
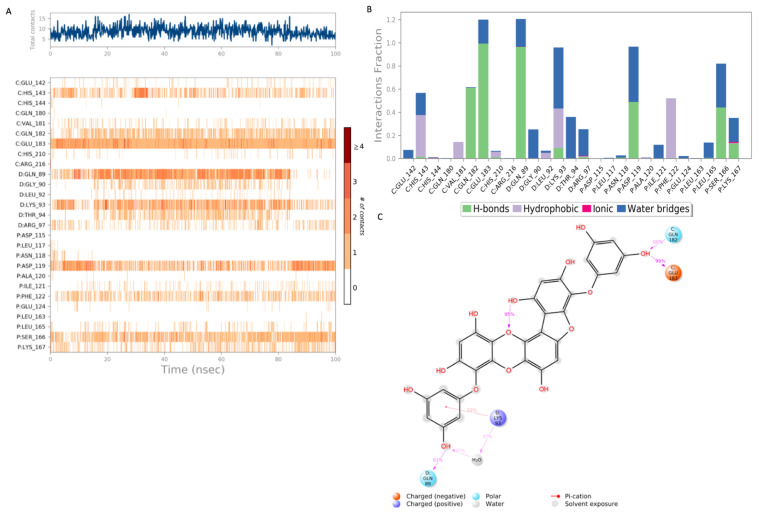
Interaction diagram of phlorofucofuroeckol with the GTPCH1-binding pocket. (**A**) Interaction of phlorofucofuroeckol with residues in each trajectory frame. The depth of color indicates the level of interaction with contact residues; (**B**) the protein–ligand contacts show the bonding interaction fractions and the nature of the interactions; (**C**) graphical 2D illustration of phlorofucofuroeckol interacting with the protein residues during MD simulation. Interactions shown occurred over 30% of the simulation time. C—chain C of the GTPCH1-binding pocket; D—chain D of the binding pocket; P—chain P of the binding pocket.

**Figure 7 ijms-24-01210-f007:**
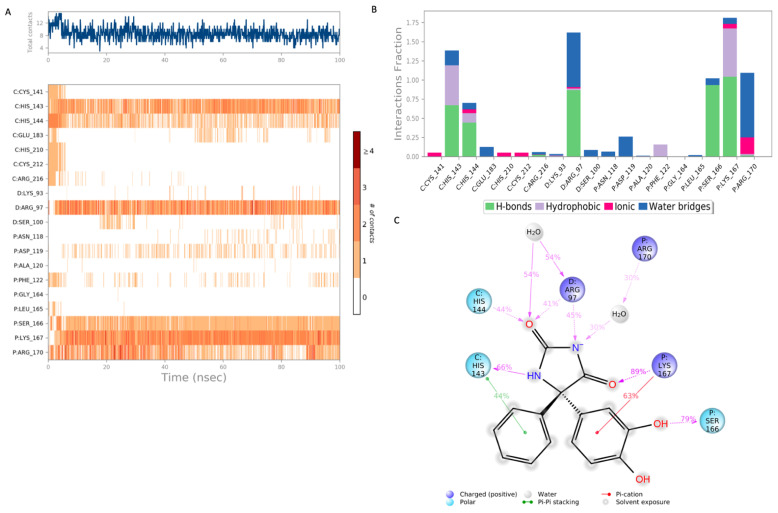
Interaction diagram of phenytoin catechol with the GTPCH1-binding pocket. (**A**) Interaction of phenytoin catechol with residues in each trajectory frame. The depth of color indicates the level of interaction with contact residues; (**B**) the protein–ligand contacts show the bonding interaction fractions and the nature of the interactions; (**C**) graphical 2D illustration of phenytoin catechol interacting with the protein residues during MD simulation. Interactions shown occurred over 30% of the simulation time. C—chain C of the GTPCH1-binding pocket; D—chain D of the binding pocket; P—chain P of the binding pocket.

**Figure 8 ijms-24-01210-f008:**
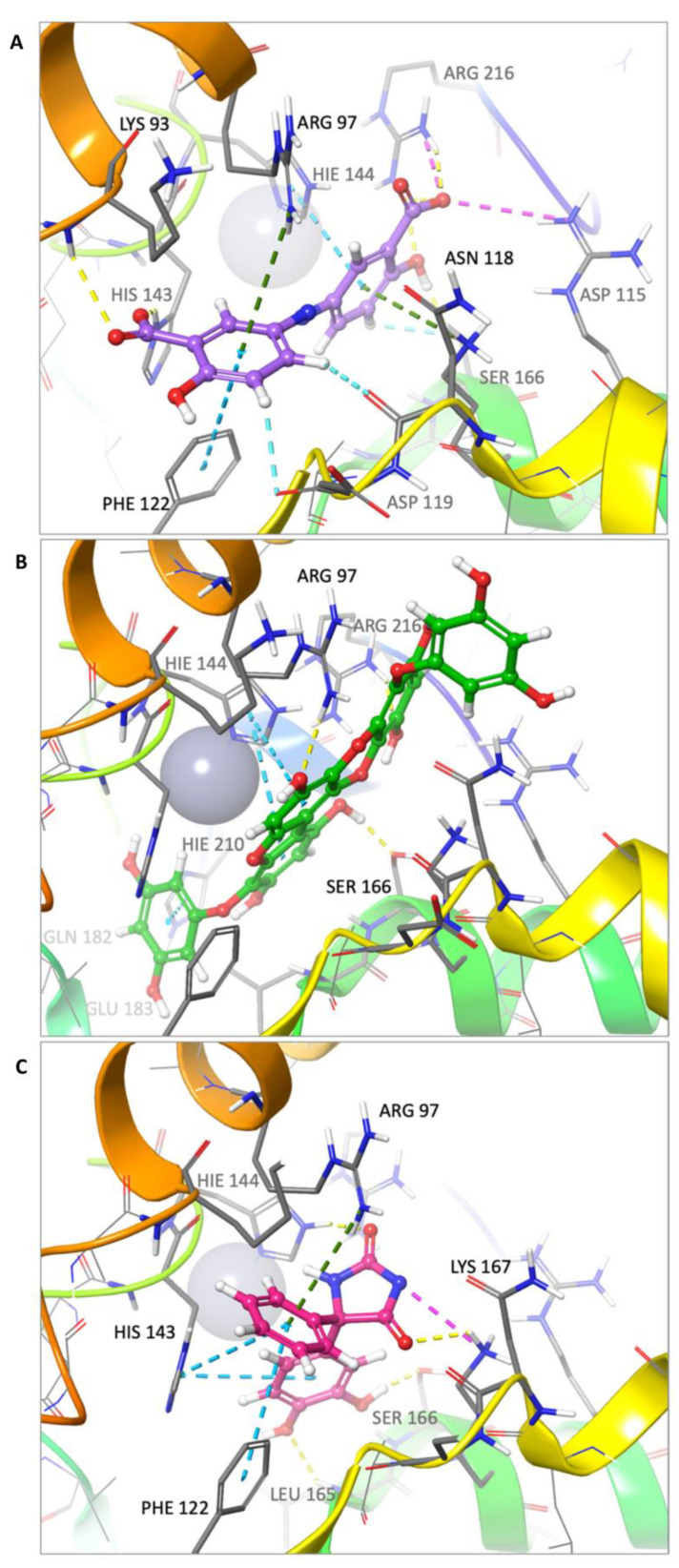
(**A**) Olsalazine, (**B**) phlorofucofuroeckol and (**C**) phenytoin catechol, docked in the GTPCH1-binding pocket. All bonds are depicted as dotted lines: hydrogen bonds—yellow; salt bridges—pink; aromatic H-bond—cyan; pi–pi stacking—blue; pi-cation—green. Binding residues are annotated, and the zinc ion is depicted as a grey sphere.

**Table 1 ijms-24-01210-t001:** Docking scores of identified hit compounds screened against the GCH1 active site.

Drug Name	Category/Pharmacological Property	Docking Score kcal/mol
Polydatin	Anti-inflammatory/antioxidant	−10.51
Fluorodeoxturidylate	Antineoplastic	−10.29
Thioxanthyllic acid	Immunosuppressant	−10.01
Phlorofucofuroeckol	Anti-inflammatory/antineoplastic	−10.02
Fosphenytoin	Antiepileptic	−9.96
Masprocol	Antineoplastic	−9.83
Penciclovir	Antiviral	−8.97
Fludarabin phosphate	Antineoplastic	−9.39
Tenofovir	Antiviral	−9.14
Eckol	Anti-inflammatory/antineoplastic	−8.59
Inosine	Nucleoside anti-inflamatory/neurorestorative	−8.45
Valganciclovir	Antiviral	−8.34
Fosfomycin	Antibiotic	−7.85
Olsalazine	Anti-inflammatory	−7.53
Phenytoin catechol	Metabolite of the antiepileptic phenytoin	−7.19
Vaborbactam	Antibiotic	−7.10

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
