# Peer review of "In Silico Investigation of the Human GTP Cyclohydrolase 1 Enzyme Reveals the Potential of Drug Repurposing Approaches towards the Discovery of Effective BH4 Therapeutics"

_ijms, 2023, doi:10.3390/ijms24021210_

Round 1

Reviewer 1 Report

In this theoretical work, a search was made among known drugs and natural compounds that can be repurposed for therapeutic effects on the human GTPCH1 enzyme using virtual screening and molecular dynamics modeling methods.

In general, the strategy of the present study seems to be quite reasonable and consistent. However, would like to draw attention to the following points:

1. The title of the article may not have been agreed upon, a recheck is required.

2. The library of compounds used for virtual screening looks very modest. What is the reason for searching for componds sets from known databases? By any similarity criteria or something else?

3. References in the manuscript are not in accordance with the MDPI standard.

4. It is necessary to describe in more detail the reasons for choosing the specific binding pocket of GTPCH1 for the docking study.

5. The text indicates that 6000 compounds were tested, while the SI lists 7074 (Table S2).

6. Has ADMET been analyzed for identified inhibitors? If not, why not?

7. There are a number of typos and inaccuracies in the text, it is necessary to proofread the text again. For example, line 146: is the grid box likely shaped like a cube with an edge length of 20 Å rather than a volume of 20 Å3?
What is the reason for choosing a different order of the significant decimal point of the docking score in the Table 1?

8. Figure 4: There is no need to overlap 7 compounds at the same time, this makes the picture indistinguishable.

9. How can one explain the fact that in section 3.2. two compounds (olsalazine and phenytoin catechol) are analyzed in detail, which have the most favorable binding profile based on molecular dynamics results, but at the same time these compounds have very mediocre docking scores?

Reviewer 2 Report

Interesting, but not completed yet: please find a collaborator who'll test the effect of predicted molecules on the enzyme, and report in vitro activity. Simple prediction without validation is not publishable as such - yes, it is important, but no, it cannot be published as such because it is simply the first step in the discovery (or not) of a new therapeutic indication. An organic chemist would, for example, not publish a paper entitled "I dissolved reagent A in DMSO", but proceed with the full synthesis, purification and characterization of the product before publication - even though dissolving reagent A is a paramount step of the workflow.  Note that I am not asking to validate the new indication clinically - that would be something only a pharmaceutical company can afford, and is also something that does depend on may other factors beyond direct binding affinity to the target. The successful in vitro binding test confirming  activity is NOT sufficient to guarantee in vivo efficacy of the repurposed drug... BUT it is sufficient to capture attention of pharma companies, and certainly both necessary and sufficient for this publication to be accepted. The first virtual step reported here seems technically sound, but ... it's just a video game, it's meaningless without experimental validation.

Another important point: language – the English is not very bad, but writing is extremely sloppy. Example (from the Abstract): “We identify a number of drugs and natural compounds with recognized anti-inflammatory, analgesic and cytotoxic effects including; [DO NOT USE ; IN ENUMERATION, IT’S : AT THE BEGINNIG, THEN COMMA] the animosalicylate [??AMINOSALICYLATE, perhaps]; olsalazine, the antiepileptic metabolite [WHAT “METABOLITE”? THIS IS NOT AN ACCEPTABLE MOLECULAR NOMENCLATURE]; phenytoin catechol, and the phlorotannins; phlorofucofuroeckol and eckol.”

Round 2

Reviewer 2 Report

Look, I am sick and tired of this game: more than HALF of the papers I receive for reviewing are like yours: applications of an existing modeling methodology to make and publish some untested predictions. Sorry, there is no point to remind me that the journals are full of this - but it's not because Homo Sapiens does this (nonsense, in my opinion) that you have to follow suite! My policy is and ever has been very clear on this topic: I only accept modeling and chemoinformatics papers if either (A) they are methodologically novel or (B) their predictions are experimentally tested. I have extensively explained why I believe that this is the right thing to do (and why my all reviewer colleagues should follow suite - not all of them do, but be assured that top scientists in chemoinformatics and in silico drug design do!)

Furthermore, I do this because I want to help YOU! Your argument that publishing this will allow some biologist to test your hypotheses is nothing but wishful thinking. I have now quite some years of career behind me - in both industrial and academic in silico drug design, and I can assure you that NEVER EVER a medicinal chemist or biologist has picked up some in silico predictions and sat down to test compounds because they were predicted active! Why? First of all - because, as you might know - science works on a project basis. Or, there is no such thing as a project to "fish for predicted actives from literature". Why not? First, because each project has typically one in silico group in support - so they'd have their own predictions to get tested, and they'd be VERY angry to hear that the chemists chose to test molecules published in IJMS rather than those they have predicted. Second, chemists and biologists are suspicious of this in silico oracles: at best, they'd like to ask their in-house modelers to redo the studies (also because they'd want to SEE the ligand in the active site interactively, rotate it at will, etc).  Third - what if some less than ethical colleague takes your publication, test the compounds, discovers that they are active and then writes a paper... claiming that he's the discoverer? Fourth - WHO on Earth do you think will read your paper and starts working on BH4, from scratch? This will never happen - you do not move overnight into new biological systems, because it takes a lot of effort to set up the biological testing platform. Therefore - the only people which **MIGHT** test your predictions are the people you cite as working on that biological system - and there are not many of them. So - WHY NOT SEND THEM AN E-MAIL, ASKING THEM TO CHECK OUT YOUR PREDICTIONS? What is so difficult about that? If they don't want - well, sorry, then you worked... for nothing! Not really for nothing: it will serve you as a lesson to first enter a collaboration with chemists and biologists, and start calculations only when you know that some experimentalist asked for them! This is the cruel fate of the computational chemists, myself included: we are slaves of experimentalists - unless we start developing and programming new technologies, where we are free to indulge in abstractness (in as far as benchmarks show our approaches to work). We are not using computers to print out predictions just for the sake of bolstering our CVs with longer publication lists (note - publications like this will barely get cited). Sorry - I need to insist: get those compounds tested, then come back.
